# Occupational Exposure to Halogenated Anaesthetic Gases in Hospitals: A Systematic Review of Methods and Techniques to Assess Air Concentration Levels

**DOI:** 10.3390/ijerph20010514

**Published:** 2022-12-28

**Authors:** Marta Keller, Andrea Cattaneo, Andrea Spinazzè, Letizia Carrozzo, Davide Campagnolo, Sabrina Rovelli, Francesca Borghi, Giacomo Fanti, Silvia Fustinoni, Mariella Carrieri, Angelo Moretto, Domenico Maria Cavallo

**Affiliations:** 1Department of Science and High Technology, University of Insubria, 22100 Como, Italy; 2IRCCS Ca’ Granda Foundation Maggiore Policlinico Hospital, 20122 Milan, Italy; 3Department of Clinical Sciences and Community Health, University of Milan, 20122 Milan, Italy; 4Department of Cardiac, Thoracic, Vascular Sciences and Public Health, University of Padua, 35122 Padova, Italy

**Keywords:** waste anaesthetic gases, hospital staff, inhaled anaesthetics, volatile compounds, operating rooms, healthcare workers

## Abstract

Objective During the induction of gaseous anaesthesia, waste anaesthetic gases (WAGs) can be released into workplace air. Occupational exposure to high levels of halogenated WAGs may lead to adverse health effects; hence, it is important to measure WAGs concentration levels to perform risk assessment and for health protection purposes. Methods A systematic review of the scientific literature was conducted on two different scientific databases (Scopus and PubMed). A total of 101 studies, focused on sevoflurane, desflurane and isoflurane exposures in hospitals, were included in this review. Key information was extracted to provide (1) a description of the study designs (e.g., monitoring methods, investigated occupational settings, anaesthetic gases in use); (2) an evaluation of time trends in the measured concentrations of considered WAGs; (3) a critical evaluation of the sampling strategies, monitoring methods and instruments used. Results Environmental monitoring was prevalent (68%) and mainly used for occupational exposure assessment during adult anaesthesia (84% of cases). Real-time techniques such as photoacoustic spectroscopy and infrared spectrophotometry were used in 58% of the studies, while off-line approaches such as active or passive sampling followed by GC-MS analysis were used less frequently (39%). Conclusions The combination of different instrumental techniques allowing the collection of data with different time resolutions was quite scarce (3%) despite the fact that this would give the opportunity to obtain reliable data for testing the compliance with 8 h occupational exposure limit values and at the same time to evaluate short-term exposures.

## 1. Introduction

The advent of modern general anaesthesia is undoubtedly one of the most important achievements of medicine because it allows safe performance of complex surgical and diagnostic procedures [1]. Nowadays, the most commonly used anaesthetic gases are halogenated gases, i.e., sevoflurane (C_4_H_3_F_7_O; CAS: 28523-86-6; 1 ppm = 8.17 mg/m^3^ at 1 atm and 25 °C), isoflurane (C_3_H_2_ClF_5_O; CAS: 26675-46-7; 1 ppm = 7.52 mg/m^3^ at 1 atm and 25 °C) and desflurane (C_3_H_2_F_6_O; CAS: 57041-67-5; 1 ppm = 6.87 mg/m^3^ at 1 atm and 25 °C) [2]. These chemicals appear initially in a liquid form and after being vaporized, volatile anaesthetics are administered via inhalation in a carrier gas (e.g., oxygen), alone or as a mixture (e.g., through mechanical ventilation, endotracheal tube, laryngeal mask airway, face mask, etc.). However, a certain amount of gases, known also as waste anaesthetic gases (WAGs), could be released or leak out and spread in the workplace (i.e., operating rooms, dental clinics and veterinary settings), thus giving rise to potential occupational exposure [3]. The emission of these gases in the atmospheres of operating rooms can be ascribed to various causes. The anaesthetic techniques used for the induction and/or maintenance of anaesthesia may play a fundamental role [4,5]. Exposure levels may depend on the type of mask worn by patients during anaesthesia [6]. In particular, face masks are frequently used in the treatment of paediatric patients and in this context a relevant release of anaesthetic gases from the face mask can be observed due to lack of cooperation of the patient [7,8,9,10,11,12]. However, even in the case of patient intubation, gas releases may occur during the medical procedures. In fact, the anaesthetic gases can be released from leaks in the anaesthesia system (e.g., from tubing, seals, gaskets, etc.) [13]. WAGs can also escape from around the patient’s endotracheal tube or laryngeal mask airway if the cuff is not properly inflated or the wrong size is used [14].

Other factors that could be related to exposure to WAGs are poor efficiency of the air removal/WAGs scavenging systems and of the room ventilation system [15,16,17,18]. Furthermore, improper anaesthetizing techniques and inappropriate behaviours can favour the release of WAGs. These include, for example, improperly connected tubes and fittings for the anaesthesia machine, turning on the anaesthetic gas before the scavenging system is active, not turning the gas off when the mask is removed from the patient’s face or removing the mask too quickly before the system has been flushed and the use of incorrect procedures for filling refillable vaporizers [11,13]. Finally, even during the patient’s extubating, there may be a release of anaesthetic gas from the patient respiratory system or from the apparatus [19].

Despite the constant search for safer anaesthetic methods, nowadays occupational exposure to anaesthetic gases still represents a significant risk within hospitals [20,21,22].

Anaesthetists, nurses, surgeons and other members of the medical personnel are professionally exposed to anaesthetic gases depending on work practices [23]: operating room personnel are generally more exposed than the personnel of other hospital wards. Further, a stratification of the occupational risk was hypothesized for healthcare professionals conventionally present in the operating room, according to the different level of exposure, with a higher risk for anaesthetists, a lower risk for surgeons and an intermediate risk level for the remaining nursing staff [12,21].

Many safety and health authorities and institutions, such as the U.K. Health and Safety Executive in the framework of the COSHH (Control of Substances Hazardous to Health) regulation, require the routine monitoring of WAGs concentrations in operating rooms to assess occupational exposure. This can be done by environmental and/or biological monitoring. Environmental monitoring is the most standardized way to assess exposures to WAGs and to determine the compliance with occupational limit values, while biological monitoring assumes importance because it provides additional information on the body burden of anaesthetic gases and early effects. However, the occupational exposure to WAGs can be measured through the use of different environmental monitoring techniques, following more or less complex monitoring protocols, during different types of operating sessions and with respect to various temporal resolutions [8,20,24,25,26].

To characterize the risk resulting from occupational exposure to WAGs, it is important to compare the results of exposure monitoring with occupational exposure limits. For volatile anaesthetics, the U.S. National Institute of Occupational Safety and Health (NIOSH) recommended exposure limit (REL) was set at 2 ppm related to a reference period of 1 h [27]. This limit value, published in 1977, is based on the ALARA (As Low As Reasonably Achievable) principle and on the assimilation of the different toxicological profiles to that of enflurane. However, desflurane, isoflurane and sevoflurane are not among the anaesthetic agents considered in the NIOSH document as they were not in clinical use in 1977 [28]. American Conference of Governmental Industrial Hygienists (ACGIH) proposed in 2022 a health-based limit value of 50 ppm for 8 h of exposure to isoflurane (the former ACGIH’s limit value of 75 ppm was withdrawn in 2021) [29]. As regards sevoflurane, currently only national non-binding limit values are available (Finland, Denmark, Israel, Norway, Poland, Sweden), varying between 5 and 10 ppm as time-weighted average (TWA) (8 h) and between 5 and 20 ppm as short-term exposure limit STEL (15 min) [30]. Sevoflurane, together with desflurane, is included among the substances currently under study by ACGIH [29]. Table 1 shows some national limit values for sevoflurane, isoflurane and desflurane. Further, it is worth mentioning that to date there are no internationally recognized biological exposure indices (BEIs) but only some suggested BEIs in the scientific literature [21,31,32,33].

To date, there is a quite large body of literature about occupational exposure to (or air contamination by) WAGs in hospitals. Despite the presence of systematic reviews on the subject [2,13,23], none of the existing studies systematically investigated the methods and the evolution of techniques available for exposure monitoring. In addition, the current lack of a health-based limit values for desflurane and sevoflurane leads to some problems in concluding the exposure assessment process and the consequent risk management. Furthermore, there are several different strategies, methodologies and instruments available for the air monitoring of WAGs, but to our knowledge, a comprehensive evaluation of pros and cons is currently missing. The present systematic review is then intended to give a contribution in filling these knowledge gaps and to take stock of the current challenges regarding the occupational exposure assessment of WAGs by environmental monitoring in hospitals.

It is necessary to specify that the present study only focused on “modern” inhalational halogenated gases such as sevoflurane, desflurane and isoflurane, as they are mostly used for their advantages (e.g., low solubility in blood, rapid anaesthetic induction and recovery, low rates of metabolism) [34,35]. The most important information retrieved form literature has been then summarized to provide (1) a description of the study design sampling and monitoring approaches (e.g., monitoring methods, investigated occupational settings, WAGs monitored); (2) an evaluation of time trends in the occupational exposure to sevoflurane, desflurane and isoflurane; and (3) an overview and a critical evaluation of WAGs sampling and monitoring methods as well as instruments and analytical approaches used in the literature for occupational exposure assessment of WAGs (with a focus on specificity, sensitivity, precision and benefits for exposure assessment purposes). This review is then specifically focused on current sampling and monitoring strategies, instruments and techniques for exposure monitoring of halogenated anaesthetics and was thought to sort through the currently available options for obtaining a reliable assessment of occupational exposure to WAGs.

Hospitals are the principal environments covered by this review, which are heterogeneous work settings that include day hospitals perioperative care units and operating theatres, with many people being potentially exposed. It is therefore important and often mandatory to measure WAGs concentrations depending on the specific activity conducted in the room [36]. Moreover, it is also necessary to identify, implement and maintain appropriate prevention measures either for patients or for healthcare personnel in operating theatres, where anaesthetic gases represent probably the most relevant chemical risk factor. It is equally important that the employees should be aware of the potential effects and be advised to take appropriate precautions.

## 2. Materials and Methods

### 2.1. Systematic Review

The outcomes from two different databases (Scopus and PubMed) were considered in this systematic review. For each database, a list of keywords was arranged in a search query, as reported in Table 2.

Papers were detected and then selected through the following inclusion and exclusion criteria and the PRISMA (Preferred Reporting Items for Systematic reviews and Meta-Analyses) criteria guidelines [37,38]. A total of 1160 papers (last search: 1 October 2021) was found (546 in PubMed and 614 papers in Scopus). Duplicates (340) were then removed, and 820 articles were reviewed. Only peer-reviewed scientific papers written in English or Italian, published after 1970 and which presented environmental and biological monitoring results of exposure to sevoflurane, isoflurane and desflurane in hospital environments or insights on measurement strategy and techniques were considered in this review. The following exclusion criteria were then applied: articles without abstract or other publications (i.e., case reports, reviews and conference papers), publications clearly off-topic (i.e., not related to exposure assessment of WAGs, e.g., human and animal toxicology, use of halogenated gases as bronchodilators, impact of gases on climate change, etc.) or concerning exclusively other anaesthetic gases (e.g., propofol, nitrous oxide or halothane). Moreover, papers reporting only on biological monitoring and related to settings other than hospitals (e.g., veterinary, dental clinics) were excluded. After this last screening, 101 papers were found to be suitable for the present review. More details can be found in Figure 1.

### 2.2. Statistical Analysis

A time trend analysis of the results of the studies of the final database was performed to compare the measured concentrations of WAGs across the last decades (i.e., 1990–1999, 2000–2009, 2010–2021). In addition, pie charts were constructed to highlight key differences in the main characteristics of exposure monitoring that emerged from the review (e.g., the most studied anaesthetic gas, the type of instrumentation most used, etc.).

## 3. Results and Discussions

In total, 101 articles dealing with WAG monitoring in hospital settings (Appendix A) were considered for the following analysis.

Sevoflurane was predominantly used for anaesthesia in hospitals, followed by isoflurane and desflurane (Figure 2a), and the most investigated hospital environments were operating rooms, followed by post-anaesthetic care units (Figure 2b). Most of these studies dealt with anaesthetic practices on adult people (84%), while only 16% reported results from paediatric surgery rooms (Figure 2c). The use of scavenging systems to mitigate occupational exposure to WAGs was limited to about one-third of cases (Figure 2d). As regards the environmental monitoring of WAGs, it was coupled with a contextual biological monitoring only in about one-third of cases (Figure 2e). It can be also noted that biological monitoring was also applied alone to assess occupational exposure, and 24 articles based on this approach were actually found but excluded from analysis for being off-topic (Figure 1). Most of these data were collected by personal monitoring (67%), while a combined personal and fixed-site monitoring approach was used only in 9% of cases (Figure 2f). Personal sampling is defined as a measurement within a hemisphere with a radius of 30 cm extending in front of a person’s face [40]. For monitoring, active and passive sampling was carried out equally (Figure 2g) and among the real-time techniques used, photoacoustic spectroscopy was the most represented ahead of infrared spectroscopic techniques (Figure 2h).

### 3.1. Halogenated WAGs and Concentrations Time Trends

Most of the studies (76%) focused on a single type of anaesthetic gas and only a limited number of studies (24%) presented the results referred to several gases (i.e., sevoflurane, isoflurane and desflurane). Among the three gases under investigation (i.e., sevoflurane, isoflurane and desflurane) the anaesthetic gas mostly studied in selected papers was sevoflurane (54%), followed by isoflurane (35%) and desflurane (11%) (Appendix A). It is worth noting that an increased number of papers were focused on sevoflurane over the years because of its increasing use. For this reason, it is important to develop health-based limit values for sevoflurane and desflurane capable of protecting workers from acute and chronic effects.

In addition to being well-known chemical risk factors, halogenated WAGs are key and current stressors of climate change. The fact that desflurane, which has the highest global warming potential among WAGs, is less used than isoflurane and sevoflurane is in line with climate-smart anaesthesia care procedures. The reason behind the increasing use of sevoflurane should be sought in the different pharmacokinetic and toxicodynamic properties of halogenated anaesthetic gases. While patients recovering from anaesthesia feel less confused when desflurane is used, sevoflurane is useful when rapid inhaled induction of anaesthesia is needed [41]. Sevoflurane is also characterized by a low solubility in blood, allowing more precision in the control of anaesthesia and a rapid induction to awakening, an advantage in paediatric anaesthesia and general recovery from anaesthesia [42,43].

The retrieved mean concentrations were segregated by decades and analysed. The median (max, min) of these values was calculated and depicted in Figure 3. This figure shows an increase of the sevoflurane concentrations in the 2000s and a raise in the median for isoflurane in the 2010s. Likewise, desflurane also shows an increase over the past decade.

#### 3.1.1. Concentrations of WAGs in Different Hospitals Areas

The analysis of previously published data also revealed different concentrations of WAGs in different hospital areas (Figure 4, Appendix A). Most of the studies (79%) were performed in operating theatres, 15% in post-anesthesia care units and only 4% in intensive care units. There are clear discrepancies between the number of environments investigated within hospitals. For example, there are only a few studies carried out in intensive care units, and in particular no studies are available for the decade 2000–2009. In these environments, anaesthetic gases are used mainly for sedation, while in operating rooms higher WAG concentrations are employed to achieve anaesthesia. On the other hand, exposure to WAGs in post-anaesthesia areas can be of interest. In fact, it is commonly found that the levels of WAGs can increase during the removal of masks or airway devices in post-anaesthetic care units [19,23]. Overall, in the operating rooms, higher mean values were measured (up to 19 ppm) with respect to the other investigated environments. In a few articles [44,45,46], the environmental monitoring was performed in anaesthesia rooms (excluded from Figure 4 due to the low number of available data). These are separate premises adjacent to operating theatres, used for the induction of anaesthesia in the UK. This is not the case in the other countries (i.e., US, Canada, Australia and most Scandinavian countries), where anaesthetic induction typically takes place in the operating room [47].

#### 3.1.2. Mitigation Techniques of WAGs in Different Hospital Areas

Almost 40% of the papers considered the presence of a scavenging system for collecting and removing WAGs from the operating room. This can be done by means of a device directly connected to the anaesthetic equipment or placed near the patient that takes all the gas that leaves the machine and directs it out of the operating room so, in principle, staff’s exposure to WAGs should be reduced. There are two types of scavenging system: active and passive. The first requires a suction system for the waste gas collection to a device while in the second the waste gases proceed passively down corrugated tubing through the room ventilation exhaust grill of the operating room. In addition, the scavenger interfaces can be closed by valves (the oldest ones) or open. It is well known that the use of the scavenging system reduces the occupational exposure to anaesthetic gases [48]. In seven studies [48,49,50,51,52,53,54] a comparison of air concentrations of WAGs with and without the scavenging system is reported, from which it emerged that the presence of the scavenging system in the operating rooms reduces on average by 71% (0.15–7 ppm and 0.32–16.4 ppm with and without scavenging system, respectively) the concentrations of anaesthetic gas in the room. As a confirmation of this issue, Figure 5 shows the median concentrations of WAGs measured in presence and in absence of the scavenging system.

As shown in the Figure 5, there was always an increase in median concentrations when the scavenging system was not used (1.0 (10.3, 0.3) vs. 0.7 (1.7, 0.1) ppm during the 1990s; 0.5 (19.0, 0.009) vs. 0.4 (7.0, 0.06) ppm during the 2000s; 1.0 (16.4, 0.14) vs. 0.8 (4.6, 0.004) ppm during the 2010s).

Furthermore, the maxima are also greater for the mean concentrations obtained without using a scavenging system. WAGs scavenging system is the primary line of defence against exposure; however, a properly designed heating, ventilation and air conditioning (HVAC) system can also help contribute to the dilution and removal of WAGs not collected by the scavenging system or that escape from leaks in the anaesthesia equipment or even resulting from poor work practices. Therefore, the efficiency of a scavenging system on the mitigation of exposures to anaesthetic gases appears to be appreciable.

However, scavenging systems should not be considered the only risk management measure for WAGs [52]. Leakage of anaesthetic from the anaesthesia station may also contribute to residual anaesthetic concentrations in air. In this regard, a risk control option can be represented by the double mask, consisting of a hard outer mask connected to an extraction tube, and a soft inner mask. There is a column between the outer and inner masks where leaking gas is suctioned and evacuated through the coupling house and the evacuation tubes. This mask, unlike others, can provide anaesthetic gas and reduce the WAG around the patient’s head [23].

### 3.2. Types and Evolution in Monitoring Techniques

Several techniques and approaches were used to investigate exposures or air contamination from halogenated gases. The monitoring techniques applied in the reviewed studies can be divided into real-time and time-integrated techniques (Appendix A) (Figure 6). The first instruments allow a simultaneous sampling and an analysis at high temporal resolution while time-integrated techniques provide an average sampling time data.

#### 3.2.1. Real-Time Monitoring

A detailed analysis of monitoring methods used for the sampling of WAGs in hospital settings revealed that the real–time approach is predominant (58% of the papers). Using these techniques, the identification of short-term transient peaks and an immediate exposure assessment became also possible. These instruments are essential to detect leak sources and control unacceptable exposures, and thus for a real-time management based on exposure data [55]. Since the NIOSH REL (2 ppm, 60 min) is currently in use although not specific and based according to the lowest level analytically detectable in the 1970s, it may be useful to adopt a real-time approach to assess short-term exposures. Furthermore, given the recent use of a combination of different gases to achieve optimal anaesthesia, the use of selective direct reading monitors can be crucial for a selective measurement of all the anaesthetic gases in the air mixture. Appendix A presents a summary of real-time instruments for anaesthetic gases used in the studies under review.

The most used real-time technique (Figure 6) is the photoacoustic spectroscopy (PAS) (51% of cases), mostly because of its solidness and user-friendliness [56]. The photoacoustic unit can be also connected to a multipoint sampler to obtain nearly simultaneous information from different areas of surgical units. As for other real-time approaches, a sample pre-treatment before measurement is not required. The resolution is in the order of 0.01 ppm, and the sampling interval is in the order of 60 s. However, PAS monitors are affected by some limitations. Among these, the sampling interval of PAS systems is usually quite large and dependent on the number of multi-point sampling locations and simultaneous monitoring of other gases [57]. So, it is possible that the staff is actually exposed to higher gas concentrations at some time-points not detected by the gas monitor (for details see [58]). Moreover, temperature, changes in ambient pressure, air humidity and the compresence of other gases and vapours such as N_2_O and alcohols may produce interference signal during monitoring [59]. The PAS is not suitable for personal sampling as it cannot be worn by the staff and placing tubing in the breathing zone of individuals would greatly affect their work activities, which can be particularly sensitive for healthcare workers. However, it appeared that PAS was used for personal sampling in most of the cases (90%), using tubes fixed on the health care personnel and connected to the photoacoustic monitor [4].

Single-beam infrared spectrophotometry (IR) is also widely used (40%) for the real-time monitoring of gases, probably for its rapidity to provide results: the estimated time resolution from a routine sample varies from seconds to a few minutes. As for PAS systems, the IR monitor mostly used in the reviewed studies (i.e., the MIRAN SapphIRe series (Thermo Electron Corporation, Waltham, MA)) weights about 10 kg, which is incompatible with personal monitoring. The measurement range is up to 30 ppm for halogenated agents and the detection limit varies according to the model, but in general it ranges between 0.01 and 0.2 ppm. Fourier transform infrared spectroscopy (FT-IR) devices (e.g., the GASMET DX-4030 (Gasmet Technologies (UK) Ltd.)) are more complex and powerful compared to the IR analysers discussed above because they can analyse all frequencies simultaneously and allow an analysis with a detection limit of 0.1 ppm [60]. The results of this review outlined that this technique was only used in a few studies [5,51,61,62]. The IR presents some limitations, such as the need to guarantee an adequate instrumental warm-up time before starting the measurement, sensitivity to pressure differences, which must accounted for, and interference with CO_2_ and water vapor [63].

In a minority of papers (5%), proton-transfer-reaction mass spectrometry (PTR-MS), an ultra-sensitive and selective real-time technique, was used [64,65,66]. This method allows the simultaneous determination of different molecular species based on their individual molecular mass [64,66], with a very high sensitivity (limit of detection <1 pptv). PTR-MS differs from other techniques (such as GC-MS) in that it has a fast time response of only seconds or less and the ability to make measurements over long periods of time [67]. However, interference from molecular species other than the specific analyte can occur as a potential source of error and should be accounted for.

Finally, two recent studies were conducted using ion mobility spectrometry (IMS), [68,69]. Many different technologies are available within this field, such as GC-IMS or pre-separation by a multi-capillary column (MCC-IMS). These techniques are characterized by a high sensitivity (detection limits in the ppt range) and reasonable portability taking in account their weight (about 15 kg). Ion mobility spectrometry provides direct and very fast detection (in the order of seconds). Among the possible limitations of IMS, competitive ion/molecule reactions that can mask the response of the analyte should be taken into consideration [70].

#### 3.2.2. Time-Integrated Sampling

In a considerable part of the selected articles (39%), anaesthetic gas concentrations were collected through time-integrated approaches allowing the collection of information about the average concentration throughout an entire work shift (about 8 h), which is well-suited for long-term exposure assessment. In such a case, sampling methods can be classified as active (44%) or passive (56%).

In general, active sampling of anaesthetic gases is performed using sorbent tubes packed with a suitable material (e.g., Anasorb 747, XAD-2 or ORBO-33) and connected to a low-flow sampling pump. Direct air sampling using collection containers (e.g., the FKV Bottle-Vac) or Nalophan bags was also used in a few cases. Active sampling works by means of a pump which, connected to the sample collector, sucks the air that passes through the absorber for WAGs collection.

Passive or diffusive sampling requires a longer sampling time than the active sampling to collect the same amount of analyte, but is characterized by ease of use and cheapness [63,71]. Specifically, the SKC VOC-Check, 3M 3500, Draeger Orsa 5, Zambelli TK200 and ISC Maugeri Radiello^®^ were used in the reviewed literature. Furthermore, diffusive sampling is particularly suitable for use in the operating room because of its very small size [21]. The sampling time varies upon the sampler type, the chemical of interest and the expected concentrations, for instance from less than 1 h to some weeks.

Either active or passive sampling implies sample collection onto collection media, which must be subsequently analysed in a laboratory, typically by gas chromatography after chemical or thermal desorption.

Chemical desorption is usually carried out with carbon disulfide. Instead, thermal desorption is a two-stage process occurring at high temperatures (100–300 °C) and using cold traps as refocusing devices. Desorption is followed by capillary gas chromatography, with mass spectrometric of flame ionization detection [72,73,74,75]. These approaches allow us to obtain the most reliable long-term exposure data for testing compliance with 8 h time-weighted average occupational exposure limit values (e.g., TLV–TWAs). Available studies outlined that gas chromatography is generally used for sample analysis (95%). However, in a few cases, samples collected in reservoir bags were analysed by infrared spectrometry [45,76].

#### 3.2.3. Real-Time vs. Time-Integrated Monitoring

In recent years, there has been an increasing use of real-time analysers (PAS and IR monitors) compared to time-integrated approaches (active and passive sampling), probably because these techniques allow immediate feedback of exposure levels as well as the identification of the work phases and practices most at risk, also making possible an immediate adjustment of incorrect risk management practices [77].

In fact, more and more portable real-time instruments have been developed and are now available on the market, which allows the acquisition of the best dataset for short-term exposure assessment (peak exposures). As an example, portable gas chromatographs can be regarded as promising techniques for real-time monitoring of gas at fixed positions in the environment [78]. This method can quantify the air concentrations of anaesthetic agents offering a simultaneous, selective and continuous monitoring of several different halogenated gases in a single analytical run. These instruments also offer modular arrangements so that various detectors can be used. The most sensitive monitor is based on mass spectrometry (GC-MS), which delivers lab-quality results in minutes.

It is worth noting that, in some cases, time-integrated and real-time approaches were combined [45,79,80], using a diffusive or active sampling for personal monitoring and a real-time monitor placed in a fixed position to measure air contamination in different operating rooms or also in different areas of the same one. In particular, real-time approaches (e.g., photoacoustic spectroscopy, ion mobility spectroscopy, infrared spectroscopy, portable gas chromatography (GC)) can be useful to investigate exposure profiles and to identify exposure peaks, which is crucial to assess short-term exposures by comparison with short-term exposure limits [81]. However, the real-time methods are generally less reliable (in terms of accuracy, sensitivity, precision and specificity to the chemical/variable of interest) if compared to reference-grade methods [82]. Overall, real-time methods are being successfully used complementary to reference monitoring, but they are not yet validated as alternative techniques for reference instruments. On the other hand, time-integrated measurements, typically based on diffusive or active sampling and following GC analysis, are recommended for a robust determination of 8 h averaged exposures and can be very useful for the a posteriori correction of real-time data to achieve better accuracy.

Thus, the combination of the two approaches offers the advantage of a simultaneous assessment of 8 h personal exposures of the staff to anaesthetic gases and of short-term contamination events for an integrated risk assessment and risk management process. In particular, the use of new generation real-time instrumentation not affected by relevant analytical interferences and capable of simultaneous detection of multiple gases is highly recommendable to obtain reliable information on real-time air contamination, while time-integrated methods are the most suitable for personal exposure assessment so far.

## 4. Overall Discussion

This systematic review provides information on occupational exposure to halogenated anaesthetic gases (sevoflurane, isoflurane and desflurane) in hospital settings. As part of the discussion of the available evidence, the following outcomes can be drawn.

The analysis of the selected studies indicates a predominance for only environmental monitoring, and in 30% of cases this was combined with biological monitoring. Although environmental monitoring is dominant, it may be useful to combine it with biomonitoring to obtain a complete picture for risk assessment purposes, including biomarkers of early effects. For this reason, it would be very important to fill this gap and identify valid biomarkers of exposure to complement environmental monitoring information to be used in the practice of occupational hygiene.

The time trend analysis revealed an increasing trend over the years in the number of articles on sevoflurane, possibly due to its increasing use linked to its advantageous properties over other halogenated anaesthetic gases.

Moreover, it emerged that operating rooms are the most studied hospital environments (79%). Nevertheless, there are specific scenarios, such as post-anaesthesia care units, in which high concentrations of WAGs can be measured. In fact, one study found that the levels of WAGs are higher when the endotracheal tube used to intubate patients is removed from the airways in the post-anesthesia care units [19,23].

The data analysed in this review were useful to confirm a mitigation effect of the scavenging systems on air concentrations of anaesthetic gases in operating theatres. However, scavenging systems should not be considered the only solution to reduce the WAGs in the room. Indeed, the type of mask and the presence of ventilation (i.e., Ambu bag) can also affect the release of anaesthetic gases into the air. Moreover, other major risk management options are represented by education and training aimed also at minimizing behavioural errors and increasing the awareness of the personnel exposed to WAGs.

Real-time instruments (i.e., photoacoustic, infrared and ion mobility spectrometers) are widely used (58%) for WAGs monitoring, thanks to their ability to provide results with high temporal frequency and therefore the ability to monitor the trend of WAGs concentrations and the presence of peak events. Time-integrated approaches are also widely applied, and best suited for a reliable and long-term exposure assessment.

## 5. Conclusions

In conclusion, despite the observation that environmental monitoring is dominant, it may be useful to combine it with biomonitoring to get a complete picture for risk assessment purposes, including biomarkers of early effects. For this reason, it would be very important to identify valid biomarkers of exposure to complement environmental monitoring information to be used in the practice of occupational hygiene. Furthermore, since sevoflurane is increasingly used in anaesthetic practice, it is important to derive health-based limit values for sevoflurane capable of protecting workers from acute and chronic effects. Based on the selected articles, real-time techniques are mostly used with sampling intervals consistent with the considered limit value (e.g., the NIOSH REL, 60 min). However, it can be also useful to measure the WAGs by a real-time analysis combined with a contextual time-integrated monitoring to improve accuracy and obtain the most reliable data for testing the compliance with 8 h occupational exposure limit values (e.g., TLV–TWAs) as well as for risk management purposes. As a general rule, it is very important to know in detail the uncertainty of exposure measurements and/or some analytical figures of merit such as the analytical specificity, precision and accuracy for a reliable identification of exposure events/patterns and for a sound quantification of health risks [83].

## Figures and Tables

**Figure 1 ijerph-20-00514-f001:**
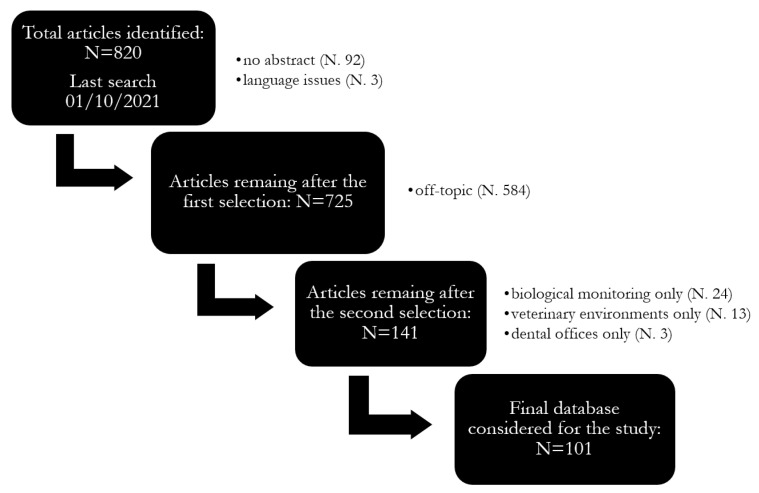
Flowchart of the literature research and review process [39].

**Figure 2 ijerph-20-00514-f002:**
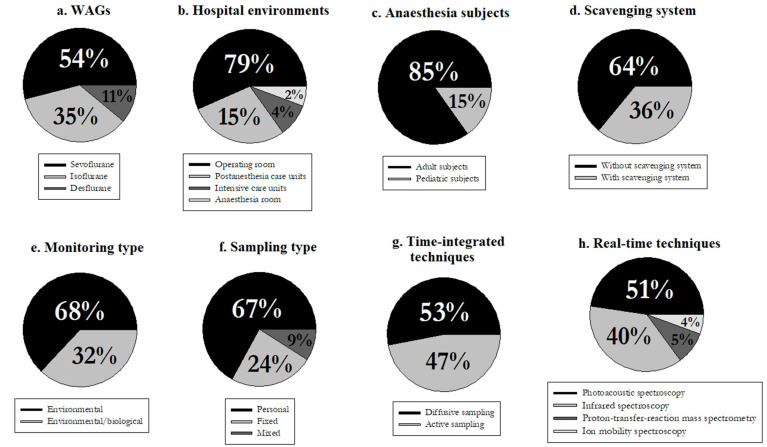
Graphical description (relative frequencies, %) of main characteristics of WAGs exposure monitoring in the investigated literature. (**a**) WAGs; (**b**) hospital environments; (**c**) anaesthesia subjects; (**d**) scavenging system; (**e**) monitoring type; (**f**) sampling type; (**g**)time-integrated techniques; (**h**) real-time techniques.

**Figure 3 ijerph-20-00514-f003:**
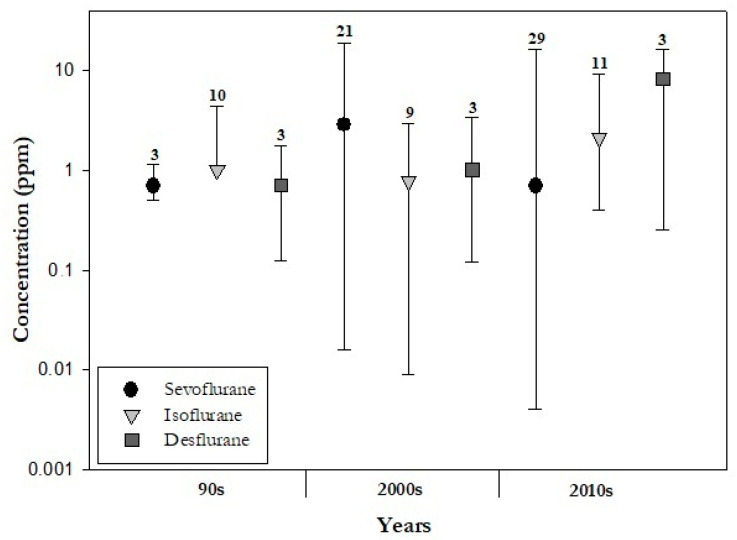
Median (circle, triangle and square), maximum and minimum (error bars) values of sevoflurane, isoflurane and desflurane concentrations in the 1990s, 2000s, 2010s, and number of studies considered per anaesthetic gas.

**Figure 4 ijerph-20-00514-f004:**
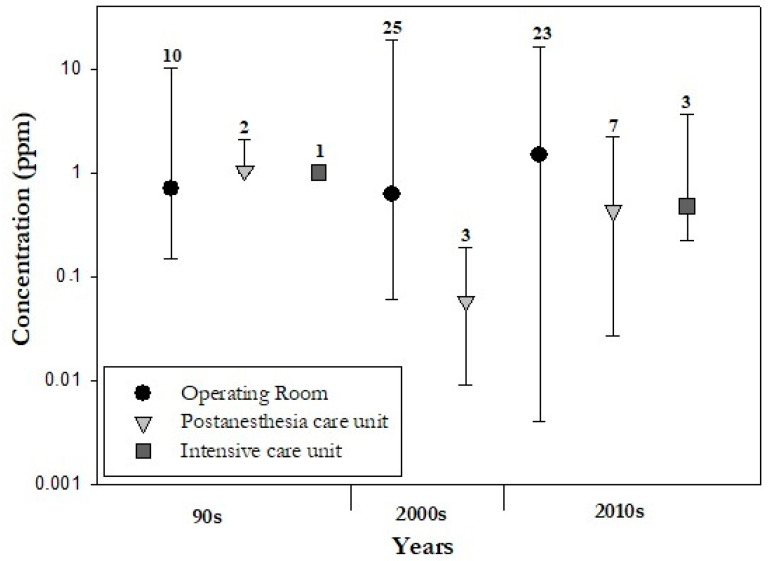
Median (circle, triangle and square), maximum and minimum (error bars) values of all three gases concentrations (sevoflurane, isoflurane and desflurane) in the operating room, post-anesthesia and intensive care units, respectively. Divided in decades (1990s, 2000s, 2010s), number of studies considered per different hospital areas are shown.

**Figure 5 ijerph-20-00514-f005:**
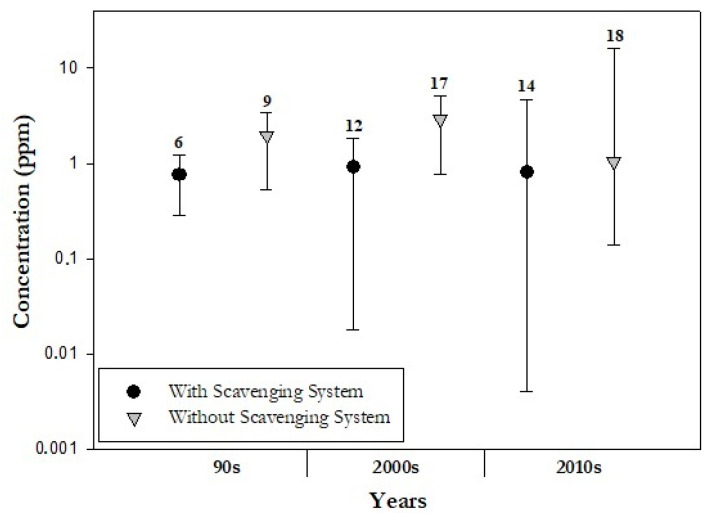
Median (circle and triangle), maximum and minimum (error bars) values of all three gases concentrations (sevoflurane, isoflurane and desflurane) with and without scavenging system, respectively. Divided in decades (1990s, 2000s, 2010s), number of studies considered per mitigation techniques of WAGs are shown.

**Figure 6 ijerph-20-00514-f006:**
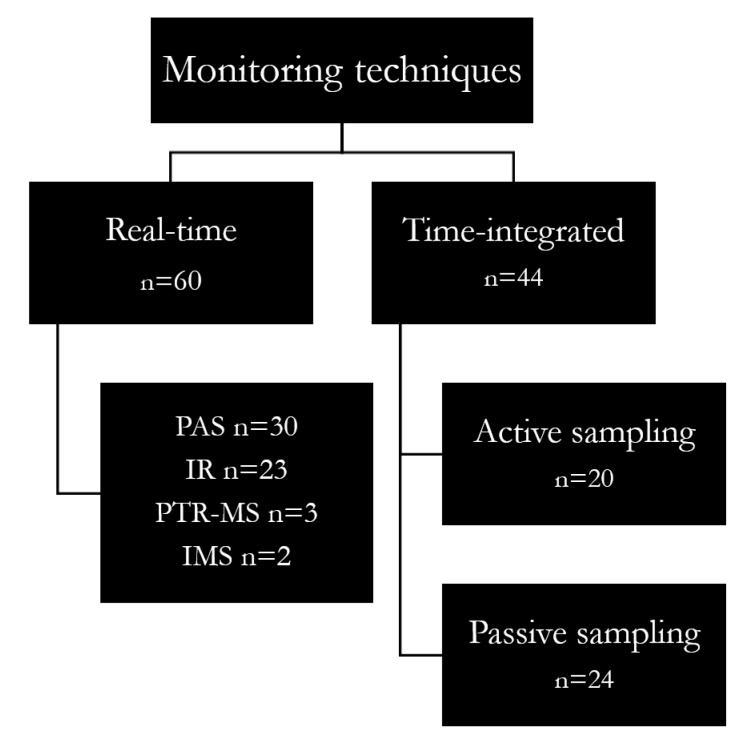
Block Diagram of monitoring approaches for anaesthetic gases. PAS = photoacoustic spectroscopy; IR = infrared spectrophotometry; PTR-MS = proton-transfer-reaction mass spectrometry; IMS = ion mobility spectrometer.

**Table 1 ijerph-20-00514-t001:** Summary of national limit values for isoflurane, desflurane and sevoflurane useful for occupational exposure assessment.

	Isoflurane				Sevoflurane				Desflurane			
	Limit Value 8 h		Limit Value Short Term		Limit Value 8 h		Limit Value Short Term		Limit Value 8 h		Limit Value Short Term	
	ppm	mg/m^3^	ppm	mg/m^3^	ppm	mg/m^3^	ppm	mg/m^3^	ppm	mg/m^3^	ppm	mg/m^3^
Austria	10	80	20	160								
Canada-Ontario	2	15										
Denmark	5	38	10 ^(1)^	76 ^(1)^					5	35	10 ^(1)^	70 ^(1)^
Finland	10	77	20 ^(1)^	150 ^(1)^	10	83	20 ^(1)^	170^(1)^	10	70	20 ^(1)^	140 ^(1)^
Ireland	50	380										
Israel	2	15	6 ^(1)^	45 ^(1)^			2 ^(1)^	16 ^(1)^				
Norway	2	15			5	35			5	35		
Poland		32				55				125		
Spain	50	383										
Sweden	10	80	20 ^(1)^	150 ^(1)^	10	80	20 ^(1)^	170 ^(1)^	10	70	20 ^(1)^	140 ^(1)^
Switzerland	10	77	80	616								
UK	50	383										

^(1)^ reference period of 15 min.

**Table 2 ijerph-20-00514-t002:** Search query arranged for each database (last search: 1 October 2021).

Database	Search Query
Scopus	(TITLE-ABS-KEY ((((occupational OR workplace OR work) AND (exposure OR monitoring) AND (sevoflurane OR isoflurane OR desflurane) OR (“anaesthetic gases”)))) AND LANGUAGE (english OR italian))
PubMed	((((occupational OR workplace OR work) AND (exposure OR monitoring) AND (sevoflurane OR isoflurane OR desflurane) OR (“anaesthetic gases”))))

## Data Availability

Not applicable.

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
