# Peer review of "Occupational Exposure to Halogenated Anaesthetic Gases in Hospitals: A Systematic Review of Methods and Techniques to Assess Air Concentration Levels"

_ijerph, 2022, doi:10.3390/ijerph20010514_

Round 1

Reviewer 1 Report

Lines 116-141: Please authors provide a table to systematically describe the exposure limits and compare the differences. The table will be more readible than the sentences in the comtent.

Figure 6: The numbers of monitoring technologies in each boxes should be indicated. Please author revise it. Additionally, the figure must be re-edited for more beautiful and dedicate.

Sec 2.2.2: Authors should further describe the statement about the instrumental analysis for the adsorbed constituents by using active and passive samplings.

Line 15: "Waste" should be "waste". 

Author Response

Lines 116-141: Please authors provide a table to systematically describe the exposure limits and compare the differences. The table will be more readible than the sentences in the comtent.

Thank you for this suggestion. We decided to move this table originally placed in the supplementary materials to the main text.

Figure 6: The numbers of monitoring technologies in each boxes should be indicated. Please author revise it. Additionally, the figure must be re-edited for more beautiful and dedicate.

The number of articles dealing with each of the monitoring methodologies is now reported in Fig. 6, whose formats were made consistent with the rest of the manuscript

Sec 2.2.2: Authors should further describe the statement about the instrumental analysis for the adsorbed constituents by using active and passive samplings.

Thanks to this useful comment, we have updated the text and added some details about chemical and thermal desorption procedures, and additional instrumental details as well (Line 494).

Line 15: "Waste" should be "waste".

OK Thanks

Reviewer 2 Report

Overall, I found this to be an informative article on an important topic. I have made a few observations and suggestions below, but I do think that the manuscript will benefit from a thorough proofreading, as they were quite a few awkward phrases. Also, whilst the focus of this systematic literature review was on occupational exposure, I wonder whether the authors should also consider environmental monitoring in the context of these gases as greenhouse gases? They did exclude papers that refer to monitoring for environmental purposes, but in my view occupational exposure and environmental effects are equally important and the same monitoring methodology will be used for both. I would also note that many of the initiatives to reduce WAGs are now down to environmental pressures.

Abstract: should the results section briefly mention the types of techniques and instrumentation that were employed?

Line 33: should this be 'despite the fact that this would…'

Line 40: should this be 'the advent of modern general anaesthesia…'

Line 41: remove the comma

line 49: vaporized

line 55: I would suggest 'thus giving rise to potential…'

Line 56: for the sentence beginning 'The dispersion…' I'm unclear as to the meaning.

Line 80: is it a scavenging system or an evacuation system (as mentioned in line 74). Need to check for consistency.

Line 89: perhaps 'significant' is preferred to 'important'

line 93: 'personnel are…'

Line 102: COSHH is a piece of legislation (regulations) rather than a health authority or institution (the relevant health authority in the UK would be the Health and Safety Executive).

Line 122: value

line 149: perhaps 'problems' rather than 'criticalities'.

Line 178: should this be 'ambulances' rather than 'ambulatories'?

Line 179: perhaps 'with many people being potentially exposed.'

Line 363: this sentence is unclear

line 406: repetition of some elements of previous sentence.

Line 412: I'm not sure that the meaning in this sentence is sufficiently clear.

Line 419: it would be interesting to know further specific technical details of the magnitude of these interferences.

Line 431: this sentence should be rewritten: the authors mean that the instrument is too heavy to allow personal measurements, though I think a similar argument was made for the PAS, yet it appeared that personal monitoring was carried out using this instrument?

Line 470: 'allowing the collection of…'

Line 510: perhaps 'which allows the acquisition of the best dataset…'

Line 531: here the authors use the term 'direct reading' and I am assuming that this is the same as 'real-time', which is used elsewhere. Please check for consistency.

Line 576: there are behavioural aspects and awareness issues that I think are also important. There have been successes in the UK NHS, where significant reductions in anaesthetic gas use has been observed. Perhaps some elements of this type of action should also be included in the discussion?

Line 590: I suggest 'despite the observation that environmental monitoring…'

Author Response

Overall, I found this to be an informative article on an important topic. I have made a few observations and suggestions below, but I do think that the manuscript will benefit from a thorough proofreading, as they were quite a few awkward phrases. Also, whilst the focus of this systematic literature review was on occupational exposure, I wonder whether the authors should also consider environmental monitoring in the context of these gases as greenhouse gases? They did exclude papers that refer to monitoring for environmental purposes, but in my view occupational exposure and environmental effects are equally important and the same monitoring methodology will be used for both. I would also note that many of the initiatives to reduce WAGs are now down to environmental pressures.

We greatly thank the reviewer for these relevant comments and suggestions. Due to the very strict deadline, we could not rely on a professional proofreading service. We have tied to revise the text to the best of our knowledge to improve language and style and hope our effort was enough. To this regard, many thanks also for all the specific recommendations about grammar and style. 

The reviewer is perfectly right about the importance of WAGs as environmental stressors with relevant impacts on climate change. This review was not focused on ambient impacts of halogenated anaesthetic gases as GHGs, so we have added a short statement (Line 270) about this key issue in the text when speaking about the relative usage of des-, iso- and sevoflurane in anaesthetic care units.

Abstract: should the results section briefly mention the types of techniques and instrumentation that were employed?

Thanks to this very useful comment, the text now includes a brief mention of the main techniques and instrumentation used for WAGs monitoring.

Line 33: should this be 'despite the fact that this would…'

OK, thanks

Line 40: should this be 'the advent of modern general anaesthesia…'

OK, thanks

Line 41: remove the comma

OK, thanks

line 49: vaporized

OK, thanks

line 55: I would suggest 'thus giving rise to potential…'

OK, thanks

Line 56: for the sentence beginning 'The dispersion…' I'm unclear as to the meaning.

We have rewritten this sentence as follows: “The emission of these gases in the atmospheres of operating rooms can be ascribed to various causes.”

Line 80: is it a scavenging system or an evacuation system (as mentioned in line 74). Need to check for consistency.

Sorry, we have inconsistently used the two words as synonyms. Only “scavenging system” was used in the new version of the manuscript

Line 89: perhaps 'significant' is preferred to 'important'

OK, thanks

line 93: 'personnel are…'

OK, thanks

Line 102: COSHH is a piece of legislation (regulations) rather than a health authority or institution (the relevant health authority in the UK would be the Health and Safety Executive).

Thank you very much: this sentence was amended to account for the fact that COSHH is a regulation and not a health authority

Line 122: value

OK, thanks

line 149: perhaps 'problems' rather than 'criticalities'.

OK, thanks

Line 178: should this be 'ambulances' rather than 'ambulatories'?

Actually, we intended ambulatory care facilities and not ambulances. The sentence was rewritten, hopefully for better clarity.

Line 179: perhaps 'with many people being potentially exposed.'

OK, thanks

Line 363: this sentence is unclear

We have tried to rephrase, hoping now the sentence is correct and clear

line 406: repetition of some elements of previous sentence.

OK, thanks

Line 412: I'm not sure that the meaning in this sentence is sufficiently clear.

We have tried to rephrase, hoping now the sentence is correct and clear

Line 419: it would be interesting to know further specific technical details of the magnitude of these interferences.

Very good point, thanks. Some technical details about cross-sensitivity of halogenated WAGs to alcoholic disinfectants can be found in Herzog-Niescery et al. 2019 (i.e. the most impacted WAG was sevoflurane and the most impacting alcohol was isopropanol) but the experimental setup was not suitable to determine the relative magnitude of these interferences. Unfortunately, we could not find anything else on the topic and remain opened to better address this topic if you think that we have missed something relevant.

Line 431: this sentence should be rewritten: the authors mean that the instrument is too heavy to allow personal measurements, though I think a similar argument was made for the PAS, yet it appeared that personal monitoring was carried out using this instrument?

We have rewritten this sentence and hope it is now clearer

Line 470: 'allowing the collection of…'

OK, thanks

Line 510: perhaps 'which allows the acquisition of the best dataset…'

OK, thanks

Line 531: here the authors use the term 'direct reading' and I am assuming that this is the same as 'real-time', which is used elsewhere. Please check for consistency.

OK, thanks. We have now used the term “real-time” consistently in the whole text.

Line 576: there are behavioural aspects and awareness issues that I think are also important. There have been successes in the UK NHS, where significant reductions in anaesthetic gas use has been observed. Perhaps some elements of this type of action should also be included in the discussion?

We totally agree with the reviewer and apologize for this forgetfulness. So, we have added a short sentence about the importance of education and training for risk management (Line 584 – 586).

Line 590: I suggest 'despite the observation that environmental monitoring…'

OK, thanks

Reviewer 3 Report

A systematic review of the scientific literature on two different scientific databases (Scopus and PubMed) was conducted about Waste Anaesthetic Gases (WAGs) which can be released into workplace air. Occupational exposure to high levels of halogenated WAGs is important to measure for determination of risk assessment and for health protection purposes. A total of 101 studies, focused on sevoflurane, desflurane and isoflurane exposures in hospitals, were included in this review. Key information is (1) study design, (2) time trend in the measured concentrations of considered WAGs, and (3) sampling strategy, monitoring method and instrument used. It is found that environmental monitoring, mainly applied for occupational exposure assessment during adult anaesthesia, was prevalent (68%), real-time techniques were used in 58% of the studies and off-line approaches were used less frequently (39%). It is concluded that the combination of different instrumental techniques allows the collection of data with different time resolutions and this would give the opportunity to obtain reliable data for testing the compliance with 8-h occupational exposure limit values and at the same time to evaluate short-term exposures.

General comments

It is the objective of this review to identify appropriate prevention measures either for patients or for healthcare personnel against WAGs in all hospital areas, where anaesthetic gases represent probably the most relevant chemical risk factor. This is required because the employees should be aware of the potential effects and be advised to take appropriate precautions. Most important papers are included in the references.

The paper addresses relevant scientific questions within the scope of the journal.

The paper presents novel concepts, ideas and tools.

The scientific methods and assumptions are valid and outlined mainly so that substantial conclusions are reached.

The results are sufficient to support the interpretations.

The description of analyses is complete and precise to allow their reproduction by fellow scientists.

The quality and information of the figures and tables are fine.

Title and abstract reflect the whole content of the paper.

The overall presentation is well structured and clear.

The mathematical symbols, abbreviations, and units are generally correctly defined and used.

Specific Comments

Paper at line 810 title is missing.

Technical corrections

References at line 632m 638, 661, 685, 737, 782, 788, 817 and 831 are incomplete.

Author Response

REV3 systematic review of the scientific literature on two different scientific databases (Scopus and PubMed) was conducted about Waste Anaesthetic Gases (WAGs) which can be released into workplace air. Occupational exposure to high levels of halogenated WAGs is important to measure for determination of risk assessment and for health protection purposes. A total of 101 studies, focused on sevoflurane, desflurane and isoflurane exposures in hospitals, were included in this review. Key information is (1) study design, (2) time trend in the measured concentrations of considered WAGs, and (3) sampling strategy, monitoring method and instrument used. It is found that environmental monitoring, mainly applied for occupational exposure assessment during adult anaesthesia, was prevalent (68%), real-time techniques were used in 58% of the studies and off-line approaches were used less frequently (39%). It is concluded that the combination of different instrumental techniques allows the collection of data with different time resolutions and this would give the opportunity to obtain reliable data for testing the compliance with 8-h occupational exposure limit values and at the same time to evaluate short-term exposures.

General comments

It is the objective of this review to identify appropriate prevention measures either for patients or for healthcare personnel against WAGs in all hospital areas, where anaesthetic gases represent probably the most relevant chemical risk factor. This is required because the employees should be aware of the potential effects and be advised to take appropriate precautions. Most important papers are included in the references.

The paper addresses relevant scientific questions within the scope of the journal.

The paper presents novel concepts, ideas and tools.

The scientific methods and assumptions are valid and outlined mainly so that substantial conclusions are reached.

The results are sufficient to support the interpretations.

The description of analyses is complete and precise to allow their reproduction by fellow scientists.

The quality and information of the figures and tables are fine.

Title and abstract reflect the whole content of the paper.

The overall presentation is well structured and clear.

The mathematical symbols, abbreviations, and units are generally correctly defined and used.

We thank the reviewer for these positive comments.

Specific Comments

Paper at line 810 title is missing.

Thank you. We have added the reference to the paper, whose details were also updated in the bibliographic section.

Technical corrections

References at line 632m 638, 661, 685, 737, 782, 788, 817 and 831 are incomplete.

All these references should be now OK. Thank you.